# Potential Application of Gambogic Acid for Retarding Renal Cyst Progression in Polycystic Kidney Disease

**DOI:** 10.3390/molecules27123837

**Published:** 2022-06-15

**Authors:** Nutchanard Khunpatee, Kanit Bhukhai, Varanuj Chatsudthipong, Chaowalit Yuajit

**Affiliations:** 1Biomedical Science Program, College of Medicine and Public Health, Ubon Ratchathani University, Sathonlamark Road, Warin Chamrap, Ubon Ratchathani 34190, Thailand; nutchanard.kh.62@ubu.ac.th; 2Department of Physiology, Faculty of Science, Mahidol University, Rama VI Road, Rajathevi, Bangkok 10400, Thailand; kanit.bhu@mahidol.ac.th; 3Research Center of Transport Proteins for Medical Innovation, Faculty of Science, Mahidol University, Rama VI Road, Rajathevi, Bangkok 10400, Thailand; varanuj.cha@mahidol.ac.th; 4College of Medicine and Public Health, Ubon Ratchathani University, Sathonlamark Road, Warin Chamrap, Ubon Ratchathani 34190, Thailand

**Keywords:** gambogic acid, MDCK cyst growth, ERK1/2, S6K, AMPK

## Abstract

Abnormal cell proliferation and accumulation of fluid-filled cysts along the nephrons in polycystic kidney disease (PKD) could lead to a decline in renal function and eventual end-stage renal disease (ESRD). Gambogic acid (GA), a xanthone compound extracted from the brownish resin of the *Garcinia hanburyi* tree, exhibits various pharmacological properties, including anti-inflammation, antioxidant, anti-proliferation, and anti-cancer activity. However, its effect on inhibiting cell proliferation in PKD is still unknown. This study aimed to determine the pharmacological effects and detailed mechanisms of GA in slowing an in vitro cyst growth model of PKD. The results showed that GA (0.25–2.5 μM) significantly retarded MDCK cyst growth and cyst formation in a dose-dependent manner, without cytotoxicity. Using the BrdU cell proliferation assay, it was found that GA (0.5–2.5 μM) suppressed MDCK and *Pkd1* mutant cell proliferation. In addition, GA (0.5–2.5 μM) strongly inhibited phosphorylation of ERK1/2 and S6K expression and upregulated the activation of phosphorylation of AMPK, both in MDCK cells and *Pkd1* mutant cells. Taken together, these findings suggested that GA could retard MDCK cyst enlargement, at least in part by inhibiting the cell proliferation pathway. GA could be a natural plant-based drug candidate for ADPKD intervention.

## 1. Introduction

Autosomal dominant polycystic kidney disease (ADPKD), a common renal genetic disorder, is usually found in adults and has an incidence of 1:1000 [1]. Cell-lining cysts along the nephrons destroy normal renal parenchymal tissue and reduce renal function. This can lead to end-stage renal disease (ERSD), which may then lead to death or the need for dialysis or kidney transplantation. Currently, tolvaptan (a V2R antagonist) is approved as a PKD intervention [2]. Therapeutic interventions for ADPKD using the anti-miR-17 miRNA family have been reported [3,4]. However, other potential drugs are still needed for effective ADPKD intervention, to provide choices in treatment.

ADPKD is caused by mutations of either the *PKD1* gene or *PKD2* gene, which encode the proteins polycystin-1 (PC1) and polycystin-2 (PC2), respectively [5]. Dysfunction of polycystin proteins can lead to increased intracellular cAMP levels, which further stimulate renal epithelial cell proliferation and increase transepithelial fluid secretion into the cyst lumen [6]. Previous studies have demonstrated that the abnormal cell proliferation is mediated through multiple intracellular signaling pathways, including the MAPK/ERK pathway [7], mTOR signaling pathway [8], and Wnt/β-catenin pathway [9]. In addition, several nutraceuticals, including steviol and curcumin, that inhibit the cell proliferation pathway, significantly slowed renal cystogenesis in cell and animal models of ADPKD [10]. Thus, regulating cell proliferation with new, natural, plant-based compounds may supply a novel perspective on the potential targets for PKD intervention.

Gambogic acid (GA, chemical formula: C_38_H_44_O_8_) is the principal bioactive and caged xanthone compound isolated from the brownish/orange gamboge resin of the *Garcinia hanburyi* tree [11]. This plant is found mainly in Cambodia, India, Thailand, Vietnam, and Hainan in China. Recent studies have reported that GA has several biological activities, such as anti-proliferative [12], anti-cancer [13], anti-metastatic [14], anti-angiogenesis [15], and anti-inflammatory activity [16], as well as induction of cell apoptosis [17]. The inhibitory effects of GA have been shown to inhibit cell proliferation, metastasis, and angiogenesis of human melanoma cells by reducing the expression of the PI3K/AKT/ERK signaling pathways [14]. In addition, GA was found to inhibit the PTEN/PI3K/AKT/mTOR pathways in esophageal squamous cell carcinoma [18]. Moreover, GA significantly inhibited cell growth and induced cell apoptosis via the inhibition of the AKT/mTORC1 pathways in U87 glioma cells [19]. It has been hypothesized that the anti-proliferative effect of GA may ameliorate cyst enlargement in an in vitro model of PKD. Therefore, the aim of the present study was to investigate the pharmacological effects and the detailed mechanisms of GA in slowing MDCK cyst enlargement and *Pkd1* mutant cell proliferation.

## 2. Materials and Methods

### 2.1. Reagents and Compounds

GA was purchased from Calbiochem (San Diego, CA, USA). It was dissolved in 100% DMSO to prepare a stock solution. Trypsin, fetal bovine serum, penicillin, streptomycin, and DMEM/F-12 Ham were purchased from Invitrogen (Carlsbad, CA, USA). Protease inhibitor cocktail was purchased from Hoffman-La Roche Ltd. (Basel, Switzerland). The BrdU cell proliferation assay kit and enhanced chemiluminescence ECL solution were obtained from Calbiochem (San Diego, CA, USA). Collagen type I (PureCol) was purchased from Advanced BioMatrix (Fremont, CA, USA). Primary antibodies (rabbit polyclonal), including anti-p-ERK1/2 (Thr202/Tyr204), anti-ERK1/2, anti-p-AMPK (Thr172), anti-AMPKα, anti-p-S6K (Thr421/Ser424), anti-S6K, anti-β-actin, and anti-rabbit IgG HRP-linked antibody, were purchased from Cell Signaling (Beverly, MA, USA).

### 2.2. Cell Cultures and Treatments

Madin-Darby canine kidney (MDCK) renal epithelial cells were kindly provided by Prof. David N. Shephard, University of Bristol, Bristol, UK, and mouse renal cystic epithelial cells *Pkd1^+^*^/*−*^ (PH2, heterozygous) and *Pkd1^−^*^/*−*^ (PN24, homozygous) cells were kindly given by Prof. Stefan Somlo, Yale University School of Medicine, Connecticut, USA. The MDCK cells were cultured at 37 °C and the *Pkd1* mutant cells were cultured at 33 °C in a 95% humidified and 5% CO_2_ atmosphere in a 1:1 DMEM/F-12 Ham medium, supplemented with 10% fetal bovine serum, 100 U/mL penicillin, 100 µg/mL streptomycin, 5 µg/mL of insulin, 5 µg/mL of transferrin, 5 µg/mL of selenium X, and 5 µg/mL interferon γ for the *Pkd1* mutant cells. The cells were trypsinized with 0.25% trypsin and centrifuged at 600× *g* before seeding.

### 2.3. MDCK Cyst Experiments

Type I MDCK cells were added to an individual well of a 24-well plate and suspended in 0.4 mL of ice-cold collagen containing 10% 10× minimum essential medium (MEM), 10 mM HEPES, 27 mM NaHCO_3_, 100 U/mL penicillin, and 10 µg/mL streptomycin, and were incubated at 37 °C for 90 min. After gelation, MDCK medium containing 10 µM forskolin was added to each well to stimulate cAMP-induced cystogenesis from day 0 to day 6, and the media was changed every two days. To evaluate the inhibitory effect of GA on MDCK cyst formation, GA at a dose of 0, 0.25, 0.5, 1, and 2.5 µM was added to the culture medium containing 10 µM forskolin from day 0. The MDCK medium containing forskolin and GA was changed every two days. On day 6, the number of cysts (with diameters ≥ 50 µm) and non-cyst (with diameters of < 50 µm) colonies were counted and captured at 10× magnification using an inverted microscope (Nikon, TE 2000-S). For MDCK cyst growth, cells were plated and sustained in collagen gel from day 0 to day 6. GA (0, 0.25, 0.5, 1, and 2.5 µM) was added to the medium containing 10 µM forskolin from day 6 to day 12. The MDCK medium containing forskolin and GA was changed every 2 days from 6 days onwards. Photographs of individual cysts (the same cysts in the collagen gel, identified by markings on the plate) were taken on days 6, 9, and 12 by an inverted microscope (Nikon, TE 2000-S). Cyst diameter (µm) was measured using Image J software. At least 30 cysts were found per condition.

### 2.4. Cell Viability Assay

The MTT assay was used to assess GA cytotoxicity. MDCK cells (20,000 cells/well) were grown in a 96-well plate for 24 h and were exposed to GA at doses of 0, 0.25, 0.5, 1, 2.5, and 5 µM in serum free media for 24 h. A 10% MTT solution (5 mg/mL) was added and maintained without light for 4 h. The medium was then removed, and 100 µL DMSO was added. The absorbance at 530 nm was measured and the percentage of cell viability was expressed as 100% of control.

### 2.5. Cell Proliferation Assay

A BrdU cell proliferation kit was used to assay the cell proliferation. MDCK and *Pkd1* mutant cells (8000 cells/well) were seeded in 96-well plates in DMEM/F-12 Ham medium supplemented with 10% FBS and grown for 24 h. Cells were treated with serum-free medium containing GA at doses of 0, 0.25, 0.5, 1, and 2.5 µM, and blasticidin (20 µg/mL) treatment was used as a positive control for 24 h. BrdU reagent solution was added 18 h later and incubated for 6 h. The absorbance was measured at 490 nm by an automated microplate reader, and BrdU cell proliferation was calculated as 100% of control.

### 2.6. Western Blot Analysis

Western blot analysis was performed as described previously [20]. MDCK, *Pkd1^+^*^/*−*^, and *Pkd1^−^*^/*−*^ treated cells with various concentrations of GA (0–2.5 µM for 24 h) were washed with PBS and lysed with ice-cold RIPA buffer (50 mM Tris-HCl, 150 mM NaCl, 1 mM EDTA, 1% Triton-X 100, 1 mM NaF, 1 mM Na_3_VO_4_, and 1 mM PMSF) containing a protease inhibitor cocktail. Samples were centrifuged at 10,000× *g* and the supernatant proteins (60 µg) were separated in 10% SDS-PAGE gels. Samples were then transferred to a nitrocellulose membrane. After blocking non-specific binding with 10% non-fat dry milk at room temperature for 1 h, membranes were incubated with primary antibodies (p-S6K, t-S6K, p-ERK1/2, t-ERK1/2, p-AMPK, AMPKα, and β-actin) overnight at 4 °C. The membranes were washed by TBS-Tween 20 solution three times, followed by incubation with secondary antibody for 1 h. After that, the membranes were washed with TBS-Tween 20 solution three times, and the band intensity of the proteins of interest was developed by chemiluminescence (ECL solution).

### 2.7. Statistical Analysis

All data are expressed as the mean ± SEM. The statistical significance of data between the treatment and control groups was determined by one-way analysis of variance (ANOVA), followed by Bonferroni’s post hoc test. Differences were determined to be statistically significant at *p* < 0.05.

## 3. Results

### 3.1. Gambogic Acid Had No Cytotoxic Effect on MDCK Cell Viability

To rule out cytotoxicity, the effect of GA (the chemical structure is shown in Figure 1a) on MDCK cell viability was determined prior to investigating the effect on MDCK cyst growth and cyst formation. Via the MTT assay, it was found that GA at doses of 0.25, 0.5, 1, and 2.5 μM did not have any effect on MDCK cell viability. However, GA at a dose of 5 μM significantly inhibited MDCK cell viability compared to the control (Figure 1b). Therefore, GA at dose of 0.25–2.5 µM was selected for further study, to determine its effect in slowing MDCK cyst growth and cyst formation.

### 3.2. Gambogic Acid Slowed MDCK Cyst Progression

To determine the effect of GA on MDCK cyst progression, experiments on MDCK cyst growth and cyst formation were performed. MDCK cells were seeded in 3D collagen gel and incubated with a medium containing 10 µM of forskolin for 6 days. MDCK cysts were treated with DMSO (control) or GA at doses of 0.25, 0.5, 1, 2.5 μM from day 6 to day 12 onwards (for the assessment of MDCK cyst growth). The result showed that GA (0.5–2.5 µM) significantly retarded MDCK cyst enlargement in a dose-dependent manner (Figure 1c,e). In addition, incubation of MDCK cysts with DMSO (control) or GA at doses of 0.25, 0.5, 1, and 2.5 μM in the presence of a medium containing 10 μM forskolin from day 0 to day 6 (for the assessment of MDCK cyst formation) showed that GA at all doses strongly inhibited MDCK cyst formation compared to the control (Figure 1d). These results suggested that GA significantly inhibits MDCK cyst progression in a dose-dependent manner, without cytotoxicity.

### 3.3. Gambogic Acid Suppressed MDCK and Pkd1 Mutant Cell Proliferation

To illustrate the effect of GA on cell proliferation in MDCK cells and mouse renal cystic epithelial cells (*Pkd1^+^*^/*−*^ and *Pkd1^−^*^/*−*^ cells), a BrdU cell proliferation assay was performed. MDCK cells were treated with DMSO (control) or GA at doses of 0.25, 0.5, 1, and 2.5 μM, and evaluated by this assay. The results showed that GA at doses of 0.5–2.5 µM significantly inhibited MDCK and *Pkd1* mutant cell proliferation in a dose-dependent manner compared to the control, as well as 20 µg/mL blasticidin as a positive control (Figure 2). These results indicated that GA retarded MDCK cyst enlargement, at least in part by suppressing the cell proliferation pathway.

### 3.4. Gambogic Acid Inhibited Phosphorylation of ERK1/2 and mTOR/S6K Protein Expression in MDCK and Pkd1 Mutant Cells

The ERK1/2 and mTOR/S6K pathways have been implicated in the proliferative response of cyst-lining cells [7,21]. To determine the mechanism by which GA suppresses the cell proliferation pathway, Western blot analysis of the phosphorylation of ERK1/2 and mTOR/S6K expression was performed. The results showed that GA (0.5–2.5 μM) significantly reduced phosphorylation of ERK1/2 and mTOR/S6K protein expression in a dose-dependent manner, both in MDCK and *Pkd1* mutant cell monolayers (Figure 3). These results indicated that GA suppresses cell proliferation through the inhibition of phosphorylation of ERK1/2 and mTOR/S6K in a dose-dependent manner.

### 3.5. Gambogic Acid Upregulated the AMPK Signaling Pathway

AMPK activators have been proposed to have therapeutic promise in PKD, and metformin was found to retard renal cystogenesis via activation of AMPK in both in vitro and in vivo models of PKD [22]. Since GA can inhibit mTOR/S6K expression, GA’s action for slowing MDCK cyst progression may involve the AMPK-dependent pathway. The effect of GA on AMPK protein expression was determined by using Western blot analysis. The result was that GA at a dose of 2.5 μM significantly elevated phosphorylation of AMPK expression, both in MDCK and *Pkd1* mutant cell monolayers (Figure 4). This result indicated that GA’s reduction of MDCK cyst enlargement may involve the AMPK-dependent pathway.

## 4. Discussion

GA was found to have an anti-proliferative effect and act as an anti-cancer agent. Previous reports have mentioned that GA inhibits PI3K/AKT/ERK signaling pathways in melanoma cells, [14] and suppresses the mTORC1 pathway through AMPK activation in glioma cells [19]. Since cell proliferation pathways such as ERK1/2 and mTOR signaling play a major role in PKD pathogenesis, in the present study, we emphasized demonstrating the inhibitory effect and a novel mechanism of the anti-proliferative effect of GA in slowing an in vitro model of PKD. The results obtained here showed that GA strongly inhibited MDCK cyst enlargement by inhibiting MDCK and *Pkd1* mutant cell proliferation, which could suppress phosphorylation of the ERK1/2 and mTOR/S6K signaling pathways, as well as activate AMP-activated protein kinase.

Presently, nutraceuticals are urgently needed for the development of new therapeutic agents in several chronic diseases. Since PKD pathophysiology involves abnormal cyst-lining epithelial cell proliferation, as well as fluid secretion pathways, a compound that inhibits these pathways could markedly retard renal cystogenesis both in an in vitro and in vivo model of PKD. Several natural compounds have been found to retard renal cystogenesis via the inhibition of cell proliferation (i.e., steviol, curcumin, ginkgolide B, chitosan, chalcone) [23,24,25,26,27]. However, other compounds with high potency, low concentration, and multitargets are still needed. GA, a xanthone compound, has various biological activities, such as anti-proliferation [12], anti-cancer [13], and anti-inflammation activities [16]. In this study, we hypothesized that GA might exert an anti-proliferative effect in slowing an in vitro cyst progression. First, we determined the effect of GA on MDCK cell viability. GA (0.25–2.5 μM) had no cytotoxic effect (Figure 1b). Next, to determine the effect of GA on cyst progression, MDCK cells were cultured in 3D collagen gel with 10 μM forskolin to mimic renal cystogenesis in an in vitro model of PKD. After treatment with GA, it was found that GA (0.25–2.5 μM) markedly retarded MDCK cyst progression (Figure 1c–e). Our previous studies showed that natural compounds such as steviol and chalcone derivative can slow MDCK cyst enlargement at a dose of 50–100 μM [24,28]. Interestingly, the effective dose of GA (0.25–2.5 μM) to retard MDCK cyst growth was very low compared to steviol and chalcone. These finding suggested that GA has higher potency for inhibiting MDCK cyst growth and cyst formation than other reported compounds.

The cAMP-dependent B-Raf/MEK/ERK signaling pathway is one of the key regulators of cell proliferation in PKD pathogenesis [6]. Previous studies have revealed that GA inhibits MAPK signaling, which inhibits cell proliferation in several types of cancer [29]. Therefore, we determined the anti-proliferative effect of GA in MDCK and *Pkd1* mutant cells by using BrdU incorporation. The results showed that GA (0.5–2.5 μM) significantly suppressed cell proliferation (Figure 2). The ERK1/2 and mTOR/S6K signaling pathways can stimulate cell proliferation in PKD [21]. To determine the detailed mechanism of GA in the cell proliferation pathway, Western blot analysis was carried out, and found that GA inhibited phosphorylation of ERK1/2 and mTOR/S6K levels in MDCK and *Pkd1* cell monolayers in a dose-dependent manner (Figure 3). This finding corelated well with previous studies showing that GA can strongly inhibit the growth of non-small-cell lung cancer by suppressing the AKT and mTOR/S6K pathways [17]. Several lines of evidence have shown that GA can inhibit the phosphorylation of ERK1/2 and JNK in a dose-dependent manner in human breast carcinoma cells [30]. GA suppressed the activation of the AKT, mTOR, and S6 pathways in ESCC cells [18]. Moreover, GA inhibited cell proliferation through the PI3K/AKT and ERK pathways in a dose-dependent manner in melanoma cells [14]. These findings suggested that GA retards MDCK cyst progression in part by inhibiting phosphorylation of ERK1/2 and mTOR/S6K signaling.

Hyperactivated ERK also affects the upregulation of the mTOR signaling pathway. mTORC1 regulates many cell processes, including cell growth, cell survival, and cell metabolism [21]. mTORC1 activation is reflected by upregulation of phosphorylation of S6, S6K, and 4E-BP1. Previously, it was found that AMPK activation attenuates renal cystogenesis in animal models of PKD by improving mitochondrial biogenesis and reducing tissue inflammation [31]. In addition, AMPK activation can retard renal cystogenesis through inhibition of mTOR-mediated cell proliferation, and AMPK activation induces significant arrest of cystic growth, both in an in vitro and ex vivo model of renal cystogenesis in ADPKD [22]. We demonstrated that GA (2.5 μM) can upregulate the AMPK level both in MDCK and *Pkd1* mutant cell monolayers (Figure 4). Previous studies have reported that chitosan oligosaccharide and chalcone derivative can retard MDCK cyst progression by activating AMPK [23,24]. To support this finding, previous studies reported that GA can inhibit glioma cells by activating AMPK, which inhibits the AKT/mTOR pathways [19]. Moreover, GA increases LKB1 expression and suppresses mTOR signaling by activating AMPK in lung cancer [32]. It is likely that GA may be involved in the activation of AMPK expression in the retarding of MDCK cyst enlargement. However, further study is still needed to elucidate the mechanism of GA in the AMPK pathway. Indeed, these findings suggested that GA’s slowing of MDCK cyst progression may involve an AMPK-dependent mechanism.

## 5. Conclusions

In conclusion, our study elucidated that GA retards MDCK cyst enlargement by inhibiting phosphorylation of the ERK1/2 and mTOR/S6K signaling pathways. Furthermore, GA significantly increases activation of phosphorylation of AMPK. Therefore, the combination treatment of an approved drug with GA could be useful for inhibiting renal cystogenesis. However, further study is still needed to investigate the inhibitory effect of GA in slowing renal cystogenesis in an animal model of PKD. Taken together, our findings suggest that GA could be a novel nutraceutical for PKD intervention in combination with other compounds.

## Figures and Tables

**Figure 1 molecules-27-03837-f001:**
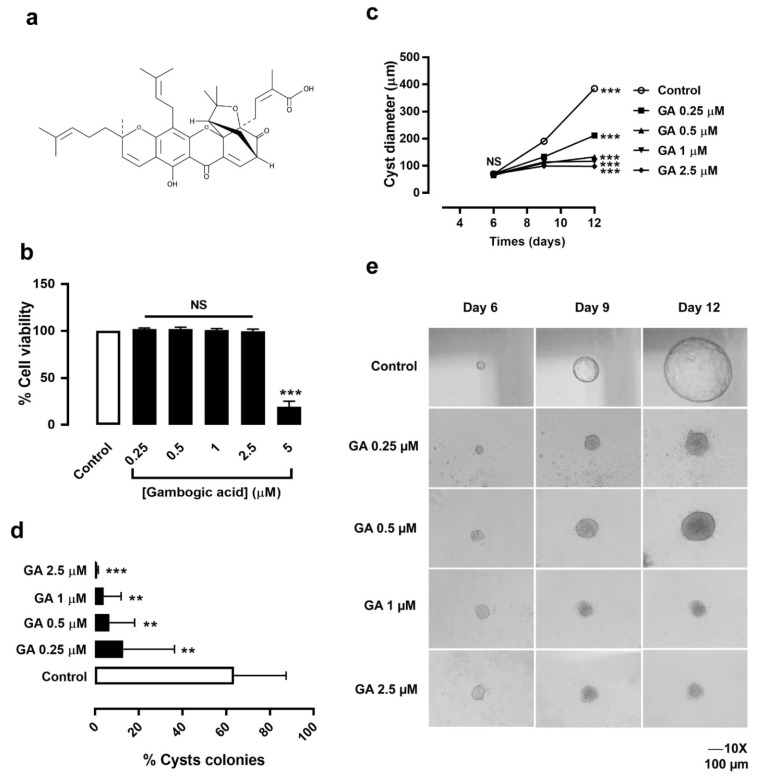
Effect of gambogic acid on MDCK cyst growth and cyst formation. (**a**) Chemical structure of GA. (**b**) MDCK cell viability after incubation with DMSO (control) and GA at doses of 0.25, 0.5, 1, 2.5, and 5 μM for 24 h (mean of percent control ± SEM, *n* = 4, *** *p* < 0.001, NS; not significant). (**c**) Inhibitory effect of GA on MDCK cyst growth. The graph shows cyst diameters at day 6, 9, and 12 after treatment with DMSO (control) and GA at doses of 0.25, 0.5, 1, and 2.5 µM from 6 days onwards (four independent experiments, mean ± SEM, *n* > 65–77 cysts per condition, *** *p* < 0.001, NS; not significant). (**d**) Inhibitory effect of GA on MDCK cyst formation. The graph shows the percent of cyst colonies at day 6 after treatment with DMSO (control) and GA at doses of 0.25, 0.5, 1, and 2.5 μM for 6 days (four independent experiments, mean ± SEM, *n* = 4 wells per condition, ** *p* < 0.01, *** *p* < 0.001). (**e**) A representative light micrograph showing MDCK cyst growth in 3D collagen gels after seeding cells for 6 days, containing 10 µM forskolin and GA at doses of 0, 0.25, 0.5, 1, and 2.5 µM for 6 days. Light micrographs were taken on the indicated days 6, 9, and 12 after cell seeding (scale bar = 100 µm and ×10 magnification).

**Figure 2 molecules-27-03837-f002:**
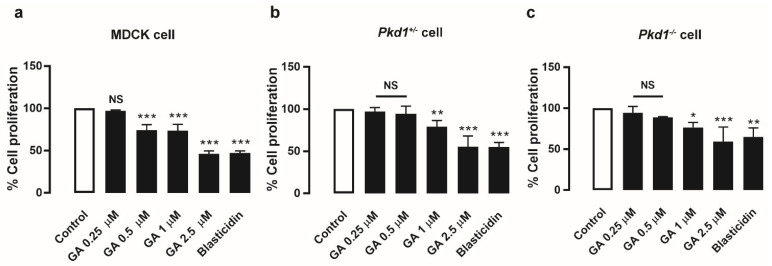
Effect of gambogic acid on cell proliferation. MDCK and *Pkd1* mutant cell proliferation was measured by BrdU incorporation. The graphs show the percentage of cell proliferation in MDCK cells (**a**) *Pkd1^+^*^/*−*^ cells (**b**) *Pkd1^−^*^/*−*^ cells (**c**) after incubation with DMSO (control), GA at doses of 0.25, 0.5, 1, and 2.5 µM, and 20 µg/mL of blasticidin for 24 h (mean of % control ± SEM, *n* = 4, * *p* < 0.05, ** *p* < 0.01, *** *p* < 0.001, NS; not significant).

**Figure 3 molecules-27-03837-f003:**
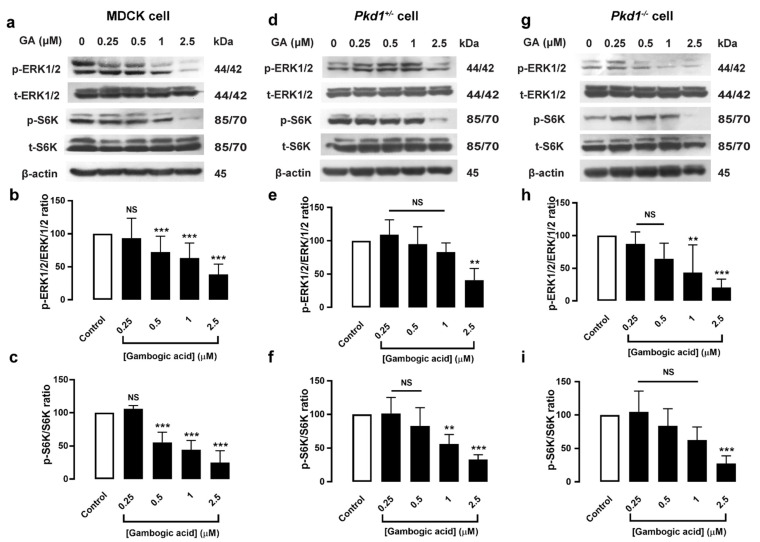
Dose-dependent effect of gambogic acid on ERK1/2 and S6K protein expression in MDCK and *Pkd1* mutant cell monolayers. Cells were seeded in a petri-dish (60 × 15 mm) and grown for 24 h. MDCK cells (**a**) *Pkd1^−^*^/*−*^ cells (**d**), and *Pkd1^+^*^/*−*^ cells (**g**) were incubated with DMSO (control) or GA at doses of 0.25, 0.5, 1, and 2.5 µM for 24 h, and blotted with the indicated antibodies. The representative band intensity of protein expression is shown. Densitometric analysis of p-ERK1/2 and p-S6K expression was normalized to β-actin and is shown as a graph of MDCK cells (**b**,**c**), *Pkd1^+^*^/*−*^ cells (**e**,**f**), and *Pkd1^−^*^/*−*^ cells (**h**,**i**), respectively (six independent experiments, mean of % control ± SEM, *n* = 6, ** *p* < 0.01, *** *p* < 0.001, NS; not significant).

**Figure 4 molecules-27-03837-f004:**
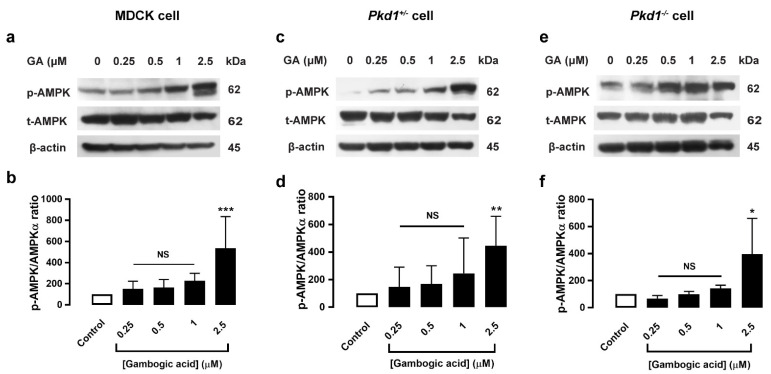
Effect of gambogic acid on AMPK protein expression in MDCK and *Pkd1* mutant cell monolayers. Cells were seeded in a petri-dish (60 × 15 mm) and grown for 24 h. MDCK cells (**a**), *Pkd1^−^*^/*−*^ cells (**c**), and *Pkd1^+^*^/*−*^ cells (**e**) were incubated with DMSO (control), or GA at doses of 0.25, 0.5, 1, and 2.5 µM for 24 h, and were blotted with the indicated antibodies. The representative band intensity of protein expression is shown. Densitometric analysis of p-AMPK/AMPKα expression was normalized to β-actin and is shown as a graph of MDCK cells (**b**), *Pkd1^+^*^/*−*^ cells (**d**), and *Pkd1^−^*^/*−*^ cells (**f**), respectively (six independent experiments, mean of % control ± SEM, *n* = 6, * *p* < 0.05, ** *p* < 0.01, *** *p* < 0.001, NS; not significant).

## Data Availability

Not applicable.

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
