# Peer review of "Potential Application of Gambogic Acid for Retarding Renal Cyst Progression in Polycystic Kidney Disease"

_molecules, 2022, doi:10.3390/molecules27123837_

Round 1
Reviewer 1 Report
The authors have described that GA could retard MDCK cyst enlargement at least in part by inhibiting the cell proliferation pathway. GA could be represented as a natural plant-based drug candidate for ADPKD intervention. The manuscript is well written however, the authors need to address the following comments to improvise the manuscript.
- In figure 1d. How many cysts were counted per condition? How many wells were there for each condition? The bar graph can be represented in the form of scatter plot to represent the variability in cyst sizes per condition. Which day is represented by the graph? Can you show the percentage of cyst colonies for 6,9 and 12 days?
2. In figure 1e. the cyst images for GA 1um and 2.5 um, the cysts are almost in the corner of the well with shadow. Replace the images with better quality images with high resolution. Do the same for all the images.
3. Did GA slowed cyst progression in Pkd1-/- compared to Pkd+/- cell lines? Any supporting evidence to show that.
4. In figure 3. d) and g) What are the protein levels of Pkd1 or Pkd2 in Pkd1+/- and Pkd-/- cell lines? How does GA affect the protein levels of Pkd1 and Pkd2? Is there any change in mTOR/p-mTOR levels after treatment with GA in Pkd1 cell lines?
5. c-Myc also plays a vital role in cell proliferation and research evidence indicates that c-Myc is up-regulated in ADPKD. Does GA downregulate c-myc expression?
6. Cite recent references for ADPKD treatment in the introduction such as Lakhia and Mishra et al. 2021, Enhancer and super-enhancer landscape in polycystic kidney disease.
Hajarinis et al. 2017, microRNA-17 family promotes polycystic kidney disease progression through modulation of mitochondrial metabolism.
Author Response
- In figure 1d. How many cysts were counted per condition? How many wells were there for each condition? The bar graph can be represented in the form of scatter plot to represent the variability in cyst sizes per condition. Which day is represented by the graph? Can you show the percentage of cyst colonies for 6, 9 and 12 days?
Reply:
The experiment for determining MDCK cyst progression is divided into 2 experiments including cyst formation and cyst growth. For cyst growth experiment, after stimulation of cyst growth in collagen gel for 6 days (day 0 – day 6), test compound was added and incubated from day 6 – day 12 and cyst diameter was measure at day 6, 9, and 12. This experiment determined an inhibitory effect of test compound in slowing cyst expansion (cyst size). For cyst formation experiment, test compound was added and incubated with cell from day 0 – day 6. The number of cyst colony (with diameter > 50 μm) was counted at day 6. This experiment determined an inhibitory effect of test compound in suppressing cyst formation (cyst number). Therefore, figure 1d represented the effect of GA on MDCK cyst formation as shown by the percentage of cyst colony 6 days after incubation with compounds.
The number of cyst and non-cyst colonies was about 70-100 cells per condition (in control group) and 20-50 cells (in treated group which depends on the potency of treated compound). Each condition in cyst formation experiment was performed in 4 wells per condition and in 4 separated experiments.
As suggested by reviewer, the bar graph can be represented in the form of scatter plot to represent the variability in cyst sizes per condition. However, in this figure (figure 1d) we would like to show the number of the cysts (the percentage of cyst colonies) after treated with GA for 6 days indicating the effect of GA in inhibiting the cyst formation. We will represent the data of cyst formation in the form of scatter plot as suggested in the future study.
In addition, based on the protocol of the cyst formation experiment, the percentage of cyst colonies was counted and shown 6 days after treatment (at day 6). However, the cyst size at day 9 and day 12 was represented in cyst growth experiment as shown in figure 1b.
The detailed of cyst formation and figure 1d was added in the revised manuscript in the figure 1 legend as shown below. (Page 7, Paragraph 1, line 170-173).
“(d) Inhibitory effect of GA on MDCK cyst formation. The graph shows the percent of cyst colonies at day 6 after treatment with DMSO (control) and GA at doses of 0.25, 0.5, 1, and 2.5 μM for 6 days (4 independent experiments, mean ± SEM, n = 4 wells per condition, **P < 0.01, ***P < 0.001).”
- In figure 1e. the cyst images for GA 1 μm and 2.5 μm, the cysts are almost in the corner of the well with shadow. Replace the images with better quality images with high resolution. Do the same for all the images.
Reply:
As suggested by reviewer, the new representative figure with better resolution of light micrographs of MDCK cyst growth in collagen gel in figure 1e (GA at doses of 1 and 2.5 μM) is now shown in the revised manuscript as suggested.
- Did GA slowed cyst progression in Pkd1-/- compared to Pkd+/- cell lines? Any supporting evidence to show that.
Reply:
The effect of GA slowed cyst progression in Pkd1 mutant cells was performed using western blot analysis to determine the detail mechanism of GA suppressing cell proliferation pathway. Our results showed that GA at dose of 2.5 μM significantly suppressed phosphorylation of ERK1/2 and phosphorylation of S6K proteins expression both in Pkd1+/- and Pkd1-/- cells. Therefore, it might suggest that the effect of GA to slow cyst progression on both Pkd1+/- and Pkd1-/- cells is quite the same as shown in figure 3d-i.
- In figure 3. d) and g) What are the protein levels of Pkd1 or Pkd2 in Pkd1+/- and Pkd-/- cell lines? How does GA affect the protein levels of Pkd1 and Pkd2? Is there any change in mTOR/p-mTOR levels after treatment with GA in Pkd1 cell lines?
Reply:
Figure 3d and g represented the effect of GA on phosphorylation of ERK1/2 and mTOR/S6K proteins expression in Pkd1 mutant cells. These cells represented the PKD1 wild-type (Pkd1+/-) and PKD1 null (Pkd1-/-). Therefore, the PC1 level in Pkd1-/- cell line might less than that of Pkd1+/- cell line [1].
Unfortunately, we did not measure the protein expression of PC1 or PC2 levels in Pkd1+/- and Pkd1-/- cell lines. Thus, the direct effect of GA on PC1 or PC2 levels was not determined either. However, we determined the effect of GA on the downstream signaling of PC1 cascade (function) such as mTOR signaling and we found that GA could inhibit mTOR/S6K protein expression both in Pkd1+/- and Pkd1-/- cells as shown in figure 3d-i.
In this study, we examined the effect of GA on the downstream signaling of mTORC1 pathway such as phosphorylation of S6 kinase. The result showed that GA at a concentration of 2.5 μM could inhibited phosphorylation of S6K protein expression both in Pkd1+/- and Pkd1-/- cells as shown in figure 3 d-i. These results indicating that GA suppressed mTOR/S6K signaling pathway in Pkd1 mutant cells.
- c-Myc also plays a vital role in cell proliferation and research evidence indicates that c-Myc is up-regulated in ADPKD. Does GA downregulate c-myc expression?
Reply:
As reviewer mentioned, c-myc plays a crucial role in cell proliferation of PKD pathogenesis and the inhibition of c-myc could slow cyst progression in animal model of ADPKD [2]. Previous study has shown that GA could downregulated c-myc mRNA expression (a downstream gene target of Wnt/β-catenin) in cholangiocarcinoma cell line [3]. In addition, GA also suppressed telomerase activity via decreasing c-myc expression in human gastric carcinoma 823 cell [4]. Therefore, it is possible that GA might inhibit c-myc expression.
In this study, we did not determine the effect of GA on c-myc expression. However, we clearly show that GA diminished cell proliferation through the inhibition of phosphorylation of ERK1/2 and phosphorylation of S6K signaling pathway. Indeed, it would be interesting to determine whether GA can inhibit c-myc expression in PKD in the future study.
- Cite recent references for ADPKD treatment in the introduction such as Lakhia and Mishra et al. 2021, Enhancer and super-enhancer landscape in polycystic kidney disease. Hajarinis et al. 2017, microRNA-17 family promotes polycystic kidney disease progression through modulation of mitochondrial metabolism.
Reply:
The references for ADPKD treatments as suggested are now added in the introduction section in the revised manuscript as shown below. (Page 2, Paragraph 1, line 38-41).
“Therapeutic interventions for ADPKD using anti-mRi-17 miRNA family have been reported [3, 4]. However, other potential drugs are still needed for effective ADPKD intervention, to give choices in treatment.”

Reviewer 2 Report
This is an original study about the use of gamboic acid to retard cyst progression. The methodology is well described and the results are preliminary and experimental but very interesting as gamboic acid could be studied as a potential drug for ADPKD.
My only concern is language, specially in the Introduction and the Discussion. I think English needs a thorough revision in these parts.
Author Response
- My only concern is language, specially in the Introduction and the Discussion. I think English needs a thorough revision in these parts.
Reply:
As suggested by reviewer, the manuscript has undergone English language editing by MDPI as shown in the revised manuscript.

Reviewer 3 Report
This manuscript "Potential application of gambogic acid for retarding renal cyst 2 progression in polycystic kidney disease" by Khunpatee et al describes a potential therapeutic strategy towards ADPKD by using Gambogic acid (GA). In this study the authors have discovered that Gambogic acid (GA) plays a pleiotropic role in kidney cystic diseases. Through a series of in vitro experiments, authors have found that GA can inhibit the phosphorylation of ERK1/2 and mTOR/S6K signaling pathways that inhibit the MDCK cyst growth. In addition GA can also increase the phosphorylation of AMPK. Authors have proposed a combination therapy of GA and dedicated drugs for PKD in future.
Firstly they found that GA upto 2.5μM does not show any cytotoxicity; and at a higher dose of 5μM, the cell viability can be compromised. Further they found that GA with a concentration of 0.5-2.5 µM successfully inhibited the MDCK and PKD1 mutant cell proliferation. Further they deciphered that in MDCK and PKD1 cell GA (0.5-2.5μM) can minimize the phosphorylation of ERK1/2 and mTOR/S6K protein. In the end they have also shown that GA are responsible for the overexpression of AMPK in both the cells as revealed by western blot.
The manuscript is very well written and easy to understand. The introduction clearly states the aim of the study. Methodology and experimental section is clearly described. Result and discussion section is very well explained.
I endorse this manuscript for acceptance.
Author Response
Reviewer #3
The manuscript is very well written and easy to understand. The introduction clearly states the aim of the study. Methodology and experimental section is clearly described. Result and discussion section is very well explained.
I endorse this manuscript for acceptance.
Reply:
Thank you very much.

Round 2
Reviewer 1 Report
The authors have addressed all the comments.